# The Continuous Adaptive Challenge Played by Arboviruses: An In Silico Approach to Identify a Possible Interplay between Conserved Viral RNA Sequences and Host RNA Binding Proteins (RBPs)

**DOI:** 10.3390/ijms241311051

**Published:** 2023-07-04

**Authors:** Massimiliano Chetta, Anna Lisa Cammarota, Margot De Marco, Nenad Bukvic, Liberato Marzullo, Alessandra Rosati

**Affiliations:** 1U.O.C. Medical and Laboratory Genetics, A.O.R.N., Cardarelli, 80131 Naples, Italy; 2Department of Medicine, Surgery and Dentistry “Schola Medica Salernitana”, University of Salerno, 84084 Baronissi, SA, Italy; acammarota@unisa.it (A.L.C.); mdemarco@unisa.it (M.D.M.); marzullo@unisa.it (L.M.); arosati@unisa.it (A.R.); 3FIBROSYS s.r.l. Academic Spin-Off, University of Salerno, 84084 Baronissi, Italy; 4Medical Genetics Section, University Hospital Consortium Corporation Polyclinics of Bari, 70124 Bari, Italy; nenad.bukvic@policlinico.ba.it

**Keywords:** bioinformatic analysis, arboviruses, vector-borne disease (VBD), RNA binding proteins (RBPs)

## Abstract

Climate change and globalization have raised the risk of vector-borne disease (VBD) introduction and spread in various European nations in recent years. In Italy, viruses carried by tropical vectors have been shown to cause viral encephalitis, one of the symptoms of arboviruses, a spectrum of viral disorders spread by arthropods such as mosquitoes and ticks. Arboviruses are currently causing alarm and attention, and the World Health Organization (WHO) has released recommendations to adopt essential measures, particularly during the hot season, to restrict the spreading of the infectious agents among breeding stocks. In this scenario, rapid analysis systems are required, because they can quickly provide information on potential virus–host interactions, the evolution of the infection, and the onset of disabling clinical symptoms, or serious illnesses. Such systems include bioinformatics approaches integrated with molecular evaluation. Viruses have co-evolved different strategies to transcribe their own genetic material, by changing the host’s transcriptional machinery, even in short periods of time. The introduction of genetic alterations, particularly in RNA viruses, results in a continuous adaptive fight against the host’s immune system. We propose an in silico pipeline method for performing a comprehensive motif analysis (including motif discovery) on entire genome sequences to uncover viral sequences that may interact with host RNA binding proteins (RBPs) by interrogating the database of known RNA binding proteins, which play important roles in RNA metabolism and biological processes. Indeed, viral RNA sequences, able to bind host RBPs, may compete with cellular RNAs, altering important metabolic processes. Our findings suggest that the proposed in silico approach could be a useful and promising tool to investigate the complex and multiform clinical manifestations of viral encephalitis, and possibly identify altered metabolic pathways as targets of pharmacological treatments and innovative therapeutic protocols.

## 1. Introduction

Despite significant advances, infectious diseases remain one of the leading causes of illness, disability, and death on a worldwide scale [1]. A multitude of variables impact the emergence and re-emergence of contagious diseases, including pathogen genetics, environmental changes, and the ever-increasing frequency of human and animal mobility, which increases the chance of interaction between viral hosts and possibly between viral host species. Among the zoonotic diseases that have ravaged people globally in recent decades there are: Acquired Immuno Deficiency Syndrome (AIDS), Ebola virus (EBOV), Severe Acute Respiratory Syndrome (SARS), Middle East Respiratory Syndrome (MERS), avian influenza, and the recently discovered COVID-19 [2]. This scenario includes zoonosis caused by arboviruses, which are transmitted by ticks and mosquitoes, frequently causing non-specific fever, and in many cases encephalitis and hemorrhages [3].

Arboviruses are RNA viruses that are classified according to structural and ecological criteria into the three most prevalent viral families (*Togaviridae*, *Flaviviridae* and *Bunyavirales*) [4].

The *Togaviridae* family is split into two genera: alphaviruses (including the Chikungunya virus) and rubiviruses (including the Rubella virus, which causes rubella) [5]. In particular, the Alphavirus genus is represented by about thirty species that are spread around the world and can infect several vertebrate species, including humans [5]. The alphavirus genome is composed of a single positive strand of RNA (+ssRNA virus) that is approximately 11–12 kb in length and is organized into two main regions: a non-structural domain located towards the 5′ end, which codes for non-structural proteins and a structural domain near the 3′ end that codes for three structural proteins [6]. Non-structural proteins are converted into one or two polyproteins before being spliced to generate four proteins known as nsP1, nsP2, nsP3, and nsP4. The structural domain instead is translated, starting from a subgenomic mRNA (26S mRNA with approximately 4100 nucleotides), into a single polyprotein. This polyprotein is processed to produce the envelope proteins (E1 and PE2, precursors of E2), a capsid protein (protein C), and two small polypeptides called E3 and 6 K. The protein C is immediately assembled with genomic RNA in the nucleocapsid, and PE2 and E1 are transported with 6 K to the plasma membrane. PE2 is then processed in the E2 protein [6,7]. The alphavirus infection has a 3–12-day incubation period and results in flu-like symptoms such as high fever, chills, headache, nausea, vomiting, and, most importantly, arthralgias, which severely restrict mobility. The fever lasts about 4 days, but it can follow a second stage of the disease, characterized by diffuse itchy maculopapular rash and fever relapse. More significant consequences are uncommon and are either hemorrhagic (within 3–5 days) or neurological, mainly in children [8].

Flavivirus is a virus with +ssRNA that belongs to the *Flaviviridae* family. Most of these viruses are classified as arthropod-borne viruses because they are transmitted through the bite of an infected hematophagous arthropod, primarily mosquitoes or ticks [9]. More than 70 species have been identified as a result of phylogenetic analyses aimed at defining the connection between various flaviviruses, classified into three groupings (clusters), and 14 serocomplexes distributed over distinct clades. The following clusters are defined through the identification of the involved vector: mosquito-borne flavivirus (MBF), tick-borne flavivirus (TBF), and no-known-vector (NKV) viruses, for which no vector has been discovered [10]. The flavivirus genome is around 10–12 kb in size and has a single open reading frame (ORF) that is translated into a polyprotein processed by cellular and viral proteases. Untranslated regions (UTRs), which are important in the replication process, flank the ORF at the 5′ and 3′ ends. The 5′ end contains a type 1 cap (m7GpppAmG), followed by a conserved stem-loop structure, but the 3′ end of the genome terminates with a conserved CUOH rather than a poly(A) tract that recognizes the site for viral RNA-dependent RNA polymerase (RdRp) [11]. Flaviviruses enter cells by endocytosis, which is mediated by mannose receptors, glycosaminoglycans, or DC-SIGN receptors (a C-type lectin receptor present on both macrophage and dendritic cell surfaces) that bind to the envelope protein E6. The low pH environment that characterizes the interior of the endosomal vesicle triggers the fusion of the viral envelope with the vesicle membrane, favoring the removal of the virus coat and the release of the genome into the cytoplasm of the infected cell [12]. Several Flavivirus species are significant human infections that can affect the homeostasis of the central nervous system (CNS). In fact, neurotropic viruses can produce neurological dysfunctions in the infected individual. These neurological conditions can advance and give rise to major inflammatory disorders that alter the CNS architecture and have a poor or even fatal prognosis. Flaviviruses include USUV, WNV, JEV (Japanese encephalitis virus), MBEV (Murray Valley encephalitis virus), and all share the neurotropism necessary to cause acute or enduring infections [13]. The virus leaves peripheral organs between the sixth and eighth day after infection, but due to its ability to cross the blood–brain barrier (BBB), it persists in the brain and spinal cord. There are different hypotheses that support the idea that flaviviruses enter the central nervous system (CNS) via the BBB. These distinct strategies include leukocytes carrying the virus across the BBB, direct virus entry after infection of brain endothelial cells, which results in impaired barrier integrity, and retrograde axonal transport-mediated virus entry after peripheral nervous system infection (i.e., olfactory nerve infection) [14].

The *Bunyavirales* family (BUNV) contains several arthropod-borne and rodent-borne viruses; it gets its name from Bunyamwera (Uganda), where the first virus was isolated from mosquitos. These viruses cause febrile diseases in humans and other vertebrates. A rodent host or arthropod vector and a vertebrate host are involved in the life cycle [15]. The majority are arboviruses (arthropod-borne viruses) that are primarily spread by arthropods (mosquitoes, ticks, sandflies). Phylogenetic analyses allowed their classification into four main genera of medical interest (*Phenuiviridae*, *Arenaviridae*, *Nairoviridae* and *Hantaviridae*), divided into 35 serogroups with more than 300 virus species and strains [16]. The entire genome is 11–12 kb in length and is made up of a single, linear molecule of negative-sense, single-stranded RNA (ssRNA-) divided into small (S), medium (M), and large (L) segments. A non-structural protein (NSs) and a nucleocapsid (N) are both encoded by the S segment on a single mRNA having overlapping open reading frames (ORFs). The NSm polypeptide and two envelope glycoproteins (G1 and G2) are encoded by the M segment and produced by the cleavage of a single polyprotein. Finally, the RNA-dependent RNA polymerase is encoded by the L segment [17].

Herein, we describe an in silico approach using different bioinformatics tools to analyze the entire genomic sequences of the main *Togaviridae*, *Flaviviridae*, and *Bunyavirales* associated with encephalopathy and identify the occurrence of specific conserved motifs capable of interacting with host proteins. Different positive and negative single-strand RNA viruses can sequester RNA binding protein (RBP) from host proteins to speed up the replication process. The host cell network could be altered by this depletion, thus interfering with nucleus-cytoplasmic traffic and causing a spatial redistribution of proteins from the nucleus to the cytoplasm.

## 2. Results

The analysis considered all of the complete sequences of the viruses currently related to the viral encephalitis phenotype. The viral genomes have been screened for possible sequences with high human similarity that are known to bind RBPs and cause disruption of specific cellular processes. In silico study of the whole genomes of *Togaviridae* (Eastern equine encephalitis virus, Western equine encephalitis virus, Venezuelan equine encephalitis, Chikungunya virus, Barmah forest virus, Ross river virus, Mucambo), *Flaviviridae* (Murray valley encephalitis virus, Usutu virus, West Nile virus, Kunjin virus, Japanese encefalite virus, Tick-borne encephalitis virus, Powassan virus, Louping ill virus, St. Louis encephalitis virus, Yellow fever, Dengue 1, 2, 3, 4, Marisma, Rocio, Zika virus, Omsk hemorrhagic fever virus), and *Bunyavirales* (culex Bunyavirus 1, La crosse) revealed three significant unique motifs. These motifs, which were found in all the viral strains investigated, are marked with a color code (red, green, and light blue) in Figure 1A and are distinguished by their ability to bind a total of 25 different RBPs. The first motif is present in all the examined strains and likely refers to an ancestral infection mechanism that survived across the species during the evolution. The other two motifs were, respectively, identified in 80% and 68% of the examined strains (Figure 1B). All the RBPs found in association with conserved RNA motifs are closely related, as shown by STRING analysis (Figure 2A); moreover, the described alterations of these proteins, due to the infective pathogenic processes, are associated with several phenotypic neurological symptoms.

### 2.1. Analysis of the First Motif (Red Motif)

The analysis revealed that the first motif has a direct interaction with proteins from the poly(A)-binding protein (PABP) family (PABPC1 (Poly(A) Binding Protein Cytoplasmic 1), PABPC4 (Poly(A) Binding Protein Cytoplasmic 4), PABPN1 (poly(A) binding protein nuclear 1), PABPC3 (Poly(A) Binding Protein Cytoplasmic 3)).

Initially, it was considered that the PABPs family protein merely protected the mRNA poly(A) tail. It is now recognized that it has a selective interaction with particular mRNA sequences and plays an important role in the metabolism of distinct mRNAs. PABPs interactions with components are involved in several physiological processes, including mRNA metabolic pathways, polyadenylation/deadenylation, mRNA export, translation, degradation, and expression regulation during development, complicate PABPs function [18]. PABPC1, PABPC4, and PABPN1 can bind the first motif, whereas PABPC3 can bind the third motif.

Through mRNA alternative polyadenylation to the 3-end of RNA, PABPC1 recruits hnRNPLL (heterogeneous nuclear ribonucleoprotein L like), which regulates the conversion of membrane Ig to secreted Ig in B cells [19]. While PABPN1 and PABPC3 are specifically necessary for progressive and efficient polymerization of poly(A) tails and are involved in cytoplasmic regulatory processes of mRNA metabolism, PABPC4 mRNA levels increase during T cell activation and regulate the stability of labile mRNA species [18,20,21].

The motif is also recognized by two SR family splicing factor proteins (SRSF2, SRSF10) that are involved in constitutive and alternative pre-mRNA splicing and are characterized by RNA recognition of the arginine/serine-rich (RS) domain [22].

Additionally, the motif also binds to proteins CNOT4 (CCR4-NOT Transcription Complex Subunit 4), HuR (ELAV Like RNA Binding Protein 1, ELAVL1), LIN28A (Lin-28 Homolog A), MATR3 (Matrin 3), PTBP1 (Polypyrimidine tract binding protein), SART3 (Spliceosome Associated Factor 3, U4/U6 Recycling Protein), TIA1 (T-Cell-Restricted Intracellular Antigen-1), U2AF2 (U2 Small Nuclear RNA Auxiliary Factor 2), ESRP2 (Epithelial Splicing Regulatory Protein 2), YBX1 (Y-Box Binding Protein 1), and YBX2 (Y-Box Binding Protein 2).

CNOT4 with insufficient E3 ubiquitin ligase activity has been associated with heart disease showing altered QT interval length. Cases of West Nile virus (WNV) encephalitis have been associated with cardiac arrhythmias, and with patients with the Ebola virus. The depletion of this protein may be related to the negative effects of several antiviral drugs that can result in *torsades de pointes* and ventricular fibrillation [23,24].

ELAVL1 is associated with CELF6 (CUGBP Elav-Like Family Member 6, also known as BRUNOL6) and binds the second motif, and both proteins are involved in the regulation of alternative splicing. ELAVL1 and CELF6 significantly affect additive control in human pathology due to their potential double depletion, which has a stronger impact than their individual depletions. In particular, CELF6 depletion results in lower brain serotonin levels, which lead to behavioral abnormalities, and destabilizes synaptic genes through mRNA interactions with 3′ UTR elements [25].

LIN28A contributes to the maturation and differentiation of neuronal stem cells. In particular, destroying dopamine neurons in the substantia nigra. LIN28A deficiency causes developmental defects and Parkinson’s disease (PD) [26]. Infection with mosquito-borne alphavirus causes selective loss of dopaminergic neurons, neuroinflammation, and widespread protein aggregation. A variety of viruses have been described with the potential for inducing or contributing to the occurrence of parkinsonism and PD [27].

MATR3, which is also bound by the third motif, is a member of a subset of RBPs that have been linked to both sporadic and familial neuromuscular disease as well as to muscular and neurodegenerative diseases such as amyotrophic lateral sclerosis (ALS) and frontotemporal dementia (FTD) [28].

PTBP1, a member of the PTB family that facilitates IRES-mediated translation and activates the replication–translation switch, is necessary for effective RNA replication. Through their recruitment in regulatory complexes during infection with several coronaviruses (CoV), DENV, and Theiler’s murine encephalomyelitis virus (TMEV), PTBP1 depletion has recently been linked to idiopathic Parkinson’s disease (iPD) [29,30].

SART3 is essential for the stabilization of complexes containing USP15, a protein that regulates NF-B activity by aiming to increase the stability of IκBα (nuclear factor of kappa light polypeptide gene enhancer in B-cells inhibitor, alpha) as well as participates in mRNA metabolism and maintains brain health. These findings suggest that the SART3-USP15 cascade disruption could result in chronic ER stress, which would speed up the neurodegenerative phenotype [31].

The ability of SART3 to bind specifically to pre-miR-34a is also very intriguing. SART3 overexpression resulted in downregulation of the miR-34a target genes *CDK4/6* and a G1 phase cell cycle arrest. The RNA-recognition motif identified in SART3 that is specific for pre-miR-34a binding supports the idea that SART3 is important for miR-34a biogenesis and might play a role in the progression of the NSCLC cell cycle. The ability of Dengue viruses to infect and replicate in human primary lung epithelium and different lung cancer cell lines is well known. The infection markedly increased the expression of IL-6 and RANTES, a chemokine mainly released by flow-dependent platelets [32].

TIA-1 protein is a transcription factor used by different viruses to support their own biology. For example, WNV produces its own RNA using TIA proteins. Furthermore, it is well known that Tick-borne encephalitis virus (TBEV) binds to viral replication sites via TIA-1 to regulate viral replication independently of the formation of stress granules (SGs) [33].

U2AF2 is thought to play a role in *IL7R* exon 6 skipping and changing the distribution of soluble IL7R isoforms on membranes during mRNA splicing and processing. The homeostatic cytokine interleukin-7 (IL-7) and the IL7R complex play a key role in the development and maintenance of T cells. The primary regulatory molecule in the IL7/IL7R axis, as well as its expression, are dynamically regulated during T cell activation and development. IL7R expression disruption contributes to immunopathologies, as demonstrated by severe immunodeficiencies, and loss-of-function variants in humans are strongly associated with risk for multiple sclerosis (MS). According to a variety of evidence, early viral infections are crucial for the development of chronic inflammatory, autoimmune, and demyelinating diseases with progressive axonal degeneration that could develop into multiple sclerosis [34].

ESRP2 does not currently appear to be connected to the occurrence of neurological disorders, although its downregulation is connected to invasive head and neck cancer and oral squamous cell carcinogenesis (OSCC). It is well known that some viruses, including the hepatitis C virus, adenoviruses, the herpes group viruses, and the human papillomavirus (HPV), have a strong correlation with oral squamous cell carcinoma [35,36].

YBX1 and YBX2 are the final two RBPs that bind the motif. It is known that the structural protein E interacts with the viral nucleocapsid of the DV through the Y-box, which is necessary for the correct formation of intracellular virus particles and their secretion [37]. YBX2 is expressed specifically in the spermatogonia to spermatocyte stage, testicular germ cells, and oocytes. ZIKV infection is widely recognized as significantly reducing spermatogenesis after producing major physiological, immunological, and endocrine damage in the testes, most likely owing to YBX2 subtraction dysregulation. Male sensitivity to flavivirus infection may be due to YBX2 expression in testicular germ cells, as evidenced by the greater incidence of antibodies in males (32.3%) compared to females [38,39].

### 2.2. Analysis of the Second Motif (Green Motif)

The motif reveals a strong binding affinity for FMR1 (Fragile X Messenger Ribonucleoprotein 1), HNRNPL (Heterogeneous Nuclear Ribonucleoprotein L), QKI (KH Domain Containing RNA Binding), SFPQ (Splicing Factor Proline and Glutamine Rich), and SNRPA (Small Nuclear Ribonucleoprotein Polypeptide A) proteins, in addition to CELF6 and PABPC3 recognized by the first motive.

Even though its function is still unknown, FMR1 has two KH domains and RGG box conserved in many RNA-binding proteins, suggesting that it is involved in RNA metabolism. It is well known that a concentration decrease in FMR1 is associated with poor learning and memory function, poor motor coordination, and poor sensorimotor adaptation [40].

Previous studies on the Japanese encephalitis virus suggested that the protein HNRNPL and the ribonucleoprotein HNRNPA2 support viral replication through the interaction with viral proteins and RNA. Reciprocal co-immunoprecipitation analyses in transfected and infected cells confirmed a specific interaction between the JEV core protein and the hnRNP proteins [41].

It has been demonstrated that QKI has a variety of functions in the regulation of viral infection, as promoting the expression of Zika proteins and, surprisingly, by inhibiting the replication of a clinical isolate-specific strain of Dengue virus (known as DENV4) through the interaction with a QKI response element (QRE). QKI deficiency increases viral infection through the suppression of the host IFNβ response following the downregulation of the mitochondrial antiviral-signaling protein (MAVS), which is essential for innate immunity response against RNA virus infection [42].

SFPQ, a specific DNA and RNA binding protein, is strictly related to neurodegenerative diseases because of its relationship with FUS protein (the neural homeostasis-binding fused in sarcoma). Frontotemporal lobar degeneration (FTLD) and amyotrophic lateral sclerosis (ALS) are genetically and clinically linked to the disruption of this interaction [43].

The last RBP able to bind the motif is SNRPA, a part of the U1 small nuclear ribonucleoprotein (U1 snRNP) complex involved in the splicing of pre-mRNAs. Even though it does not appear correlated to a neurological defect, SNRPA is highly expressed in lung adenocarcinoma (LUAD) and lung squamous cell carcinoma tissue (LUSC), as well as the progression of gastric cancer (GC) [44]. The possible removal of SNRPA by the green motif could be a therapeutic target for these kinds of tumors.

### 2.3. Analysis of the Third Motif (Light Blue Motif)

Finally, in addition to MATR3, RBM6 (RNA Binding Motif Protein 6) and RBM24 (RNA Binding Motif Protein 24), two proteins that are both highly expressed in human brain tissue, can bind the last motif. While aberrant expression of RBM6 has been involved in the development of human malignancies (i.e., growth and progression in laryngo carcinoma) as reported for SART3, SNRPA, ESRP2, and RBM24, it also plays a specific role in the differentiation of myoblast and into molecular pathways related to the expression of myogenic factors and muscle functional proteins during regeneration [45,46].

### 2.4. Enriched Analysis

To find further connections between the RBPs involved in specific molecular pathways, additional analysis was carried out. We found the presence of a subcluster that emphasizes how the dysregulation of these proteins can be linked to cancer, especially leukemia, using the STRING function of K-means cluster analysis (Figure 2B) with seven subclusters as a parameter (equivalent to a third of the number of investigated proteins). This evidence was reported in a study of 12,573 dengue patients, where stratified analyses by different follow-up times demonstrated that the risk of leukemia was considerably increased only between 3- and 6-years following dengue virus infection. This finding also supports the evidence that the dysregulation of RBM6, SART3, ESRP2, and SNRPA is linked to several tumor types.

Finally, the same list of RBPs was also investigated using Enrichr (available at: https://maayanlab.cloud/Enrichr/, accessed on 16 April 2023), a software that allows the simultaneous run of multiple searches in various databases. In addition to the expected result confirming the association of the proteins with RNA metabolism—as reported in the top 4 terms enriched in the Human KEGG (Kyoto Encyclopedia of Genes and Genomes database) and in the top 10 terms enriched in the “GO Biological Process” database in DisGeNET (a discovery platform encompassing one of the largest publicly available collections of genes, proteins, and variants relevant to human disease) (Figure 3A,B) RBPs alterations are mainly related to neuropathy and weakening, as shown in the UMAP scatterplot (Figure 3C).

This additional evidence confirms that the host cell network is altered by the depletion or dysregulation of RBPs, which affects nucleus-cytoplasmic traffic and the spatial redistribution of proteins from the nucleus to the cytoplasm, resulting in the onset of clinical symptoms, and neurological disorders.

## 3. Discussion

Infectious diseases continue to rank among the top global causes of illness, disability, and death despite significant advancements in the treatment of viral infections [1]. Numerous factors, including environmental changes, the genetics of the pathogens, and the increased frequency of animal and human movement, which increases the possibility of contact between hosts and potential host species, can cause contagious diseases to emerge and recur [1]. Anthropogenic alteration of areas with high biodiversity has produced a variety of hotspots where the risk of zoonosis is increasing. These hotspots are caused by the creation of new areas of contact that involve human structures, natural areas, and possibly new infections [1,47]. Data on emerging infectious diseases that have plagued the human population over the last three decades show that 75% are the result of a pathogen being transferred from animals, particularly wild animals, to humans. In this spillover process the pathogen evolves and gains the ability to infect, replicate, and spread across other species, including humans [47,48]. Such situations are also made worse by intensive farming, where high animal density and low genetic diversity create a favorable environment for pathogen spread, resulting in increased interactions between humans, animals, and wildlife, as well as the possibility of breeding farm animals into intermediate hosts, facilitating pathogen transmission [48,49]. Another aspect of zoonoses and infections in general is related to the concept of viral *quasispecies*, which describes an error-prone replication, and demonstrates a sophisticated replication adaptive system in response to environmental stimuli [50]. Similar to how the immune system of vertebrates expands clonally in response to antigenic stimuli, viral *quasispecies* also benefit from a molecular memory based on the existence of a dynamic population of complex mutant genomes [51]. This determines the coexistence in the host of a primary sequence (dominant nucleotide sequence) and a range of mutant sequences distinguished by the set of copy errors related to the virus’s capacity for replication [52].The highest mutation rate among living species is found in RNA viruses (between 10–3 and 10–5 errors per nucleotide and replication cycle), followed by retroviruses (which have extremely high mutation rates and exist as complex genetically heterogeneous populations) and DNA viruses (10–8 to 10–6 substitutions per replication cycle) [53]. Both the primary sequences and the mutant spectra are extremely short during RNA virus infections because environmental changes or, in the case of SARS-CoV2, the potential use of vaccines directed against a single protein, can frequently upset the population balance of viral genomes [54]. In addition to functioning as an essential adaptive strategy, the genetic organization of *quasispecies* has a range of biological effects, some of which are directly related to viral persistence but are not always associated with infection [51]. Infection results from an interaction between the virus, the host, and/or the environment, and can take one of two different forms: acute or persistent [55]. The acute infection strategy allows for a transient infection in which the host’s immune response only attends to eliminate or prevent the continuation of the infection in the same host, following the succession of replicative cycles of the virus. To continue the infectious cycle, viruses that belong to this category (i.e., influenza, rhinovirus, and SARS-CoV-2) need to find a new host during the short window of replication. Contrarily, virus persistence in a host occurs after an initial phase of replicative infection and the host’s antiviral response, during which the virus continues to have the capacity to replicate continuously or irregularly in the same host for a predetermined amount of time. The host immune response does not completely eradicate these viruses [55]. The ability of the virus to survive the host immune response requires enough susceptible cells replicating at the same rate as the virus, and the presence of a latent condition in which the replicative activity of the virus may be partially or completely suppressed for prolonged periods while retaining the ability to reactivate; these are all requirements for persistence in an organism [56]. In this scenario of a complicated and ongoing adaptive fight between the host and virus, it is critical to develop quick analysis methods that can pinpoint the fundamental causes of infection and implement possible treatments.

The suggested analytical pipeline revealed that the mere presence of the “*Togaviridae*, *Flaviviridae*, and *Bunyavirales*” genomes in the host cell could predict the depletion of particular RBPs and that the depletion of these proteins could change metabolic pathways related to the clinical phenotype. Different positive and negative single-strand RNA viruses can sequester RBPs from host proteins to speed up the replication process, disrupt nucleus-cytoplasmic traffic, and lead to a spatial redistribution of proteins from the nucleus to the cytoplasm, altering the host cell network [57,58]. It has been demonstrated via individual RBP analysis that dysregulation is related to clinical manifestations such as neuropathy, weakness, and, in severe cases, encephalitis by infecting host neurons. The severity of the virus’s effects depends on its virulence and the maturity of the infected neuron. Additionally, this outcome was attained by utilizing the Enrichr software and a DisGeNET database query, which directly relates the dysregulation of these proteins with clinical manifestations. The investigation also discovered that all strains had the conserved first motif, which is most likely the source of the more sophisticated and ancient molecular mechanism of infection. The three motifs, which have a specific role in the regulation of viral RNA maturation, can be exploited to develop compounds that limit the removal of these RBPs, hence inhibiting infection. We also highlighted that the alteration of these proteins is related to distinct cancers, especially leukemia, due to the subcluster enrichment analysis using STRING. The emergence of malignancies has been linked in the literature to alphaviruses or flaviviruses, e.g., dengue infection.

## 4. Materials and Methods

The analysis pipeline, which consists of four major steps, was carried out using several online bioinformatics tools. The in silico approach was applied on the whole known genomic sequences of the main *Togaviridae*, *Flaviviridae*, and *Bunyavirales* strains related to encephalopathy to assess the presence of specific conserved motifs capable of interacting with host proteins. MEME-ChIP, which performs a comprehensive motif analysis (including motif discovery) on large sets of sequences identified by ChIP-seq or CLIP-seq experiments on Human DNA (http://meme-suite.org/tools/meme-chip, accessed on 11 February 2023), was used to analyze the entire genomic sequences of the main *Togaviridae*, *Flaviviridae*, and *Bunyavirales* strains [59]. All identified motifs were used as queries for Tomtom (http://meme-suite.org/doc/tomtom.html, accessed on 11 February 2023), another MEME suite tool that compared the motifs to a database containing a curated and nonredundant collection of experimentally discovered and proven RNA binding site proteins on the human genome. Using the Benjamini and Hochberg method, Tomtom calculated the q-value, which is the minimal false discovery rate at which the observed similarity would be considered significant [60]. A list of human RBPs that recognize the common conserved domain distributed on viral genomes was obtained for all motif queries (Figure 1). STRING v11.5 (available at: https://string-db.org/, accessed on 12 February 2023), another software tool, was used to identify the strongest related correlation within the query RBPs set using a guilt-by-association approach. The bioinformatics tool drew on a large database of functional interaction networks from various organisms, and each related RBP can be traced back to the source network that was used to make the prediction [61].

The entire list of RBPs was used as input for Enrichr (https://maayanlab.cloud/Enrichr/, accessed on 16 April 2023), a web-software application that integrates different methods for ranking enriched terms, and various interactive visualization tools to show enrichment results using the JavaScript library “Data Driven Documents (D3)”. In addition, the software provides various visual summaries of the collective functions of gene lists [62].

## 5. Conclusions

The analysis provides new scenarios for the potential use of these viruses as an alternative therapeutic strategy in cancer patients. It is generally known that in pre-clinical investigations against cancer, recombinant vaccines based on alphaviruses have both preventive and therapeutic efficacy [63]. For instance, the potential barring of the discovered motifs that can bind RBPs could restore the proper availability of the proteins, reducing both the clinical symptoms and, indirectly, the risk of forming tumors. In conclusion, the suggested pipeline is simple to use, and this easily repeatable method could be used not only to comprehend the mechanism of infection and, consequently, the characterization of the RBPs involved, but also to identify metabolic networks suitable to identify tissue specific biomarkers and potential pathway dysfunction, which are helpful for developing a potential vaccine or therapeutic approach. The implementation of these integrated pipelines for analysis, which can be created in a single user-friendly interface, could support infection investigations by offering quick access to information on the analysis of multiple datasets (such as genotype and transcriptome data), particularly during times of emergency. Obviously, interindividual genetic variations may contribute to some of the observed variability in phenotypic responses but identifying these regulatory factors should improve the diagnostic sensitivity and accuracy of cohort classification and, as a consequence, facilitate therapy.

## Figures and Tables

**Figure 1 ijms-24-11051-f001:**
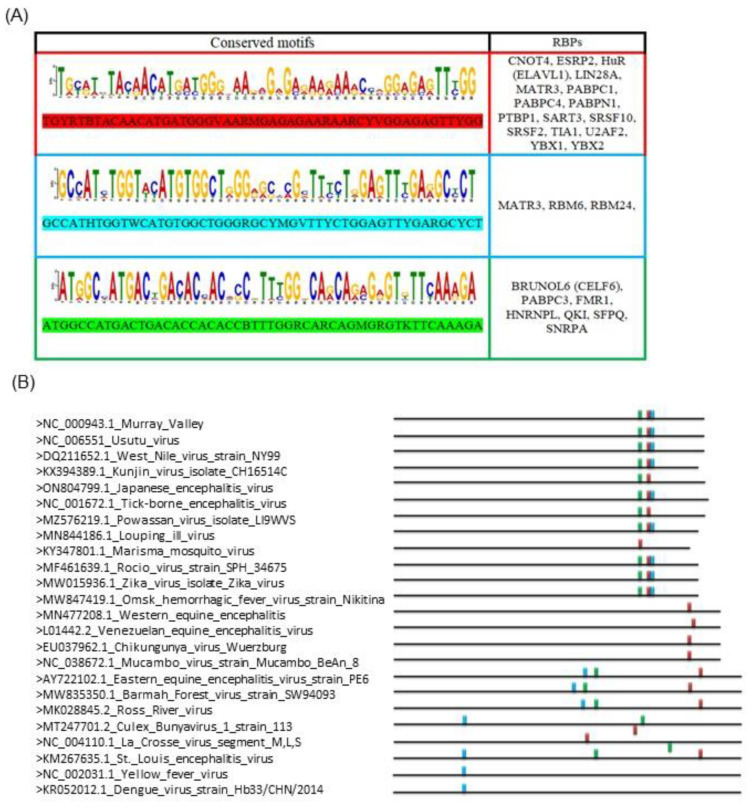
(**A**) The conserved sequences that were found by comparing all the strains are included in this figure. All consensus motifs are reported using “motif logos” and according to IUPAC nomenclature. Moreover, a complete list of RBPs has been supplied. (**B**) RBPs binding site distribution on arbovirus sequences analyzed. The colors red, green, and light blue correspond to the motifs shown in (**A**).

**Figure 2 ijms-24-11051-f002:**
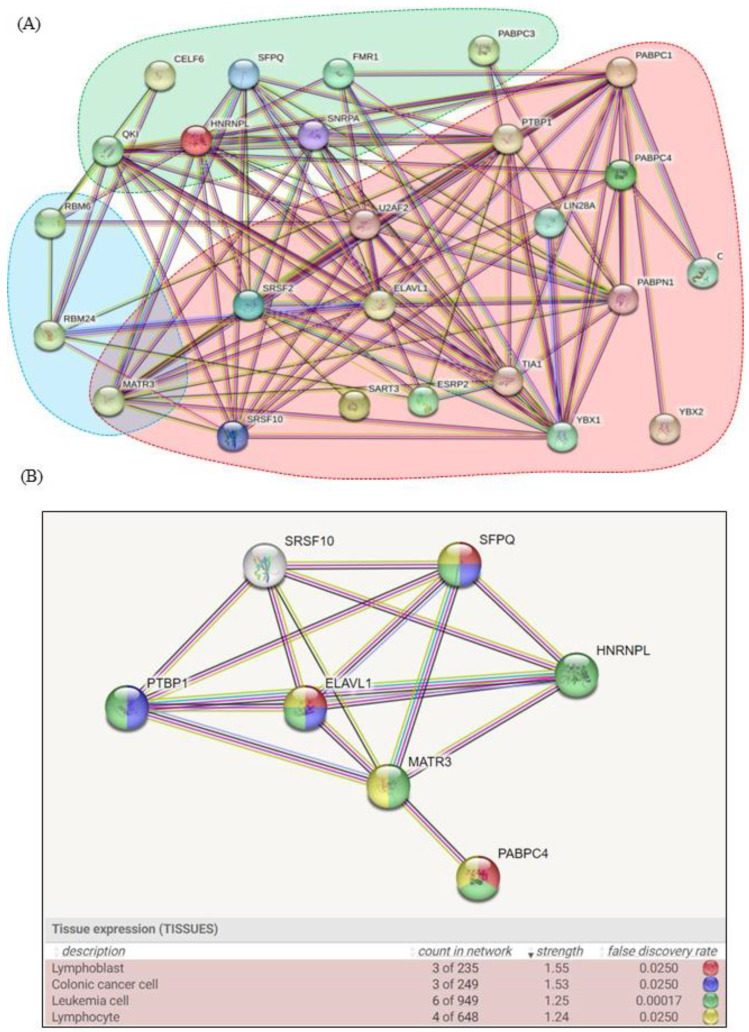
(**A**) The graph displays the connection between all known RBPs, which are grouped by the motif to which they can bind. (**B**) The set of proteins in the network is provided in the table under “Count in Network”. Strength (observed/expected: Log10) indicates the size of the enrichment effect. It is the ratio between the number of proteins in query that are annotated with a term and the number of proteins we would anticipate in a randomly generated network of the same size. False discovery rate indicates the importance of the enrichment. The Benjamini–Hochberg method is used to adjust the *p*-values for multiple testing within each category and is shown.

**Figure 3 ijms-24-11051-f003:**
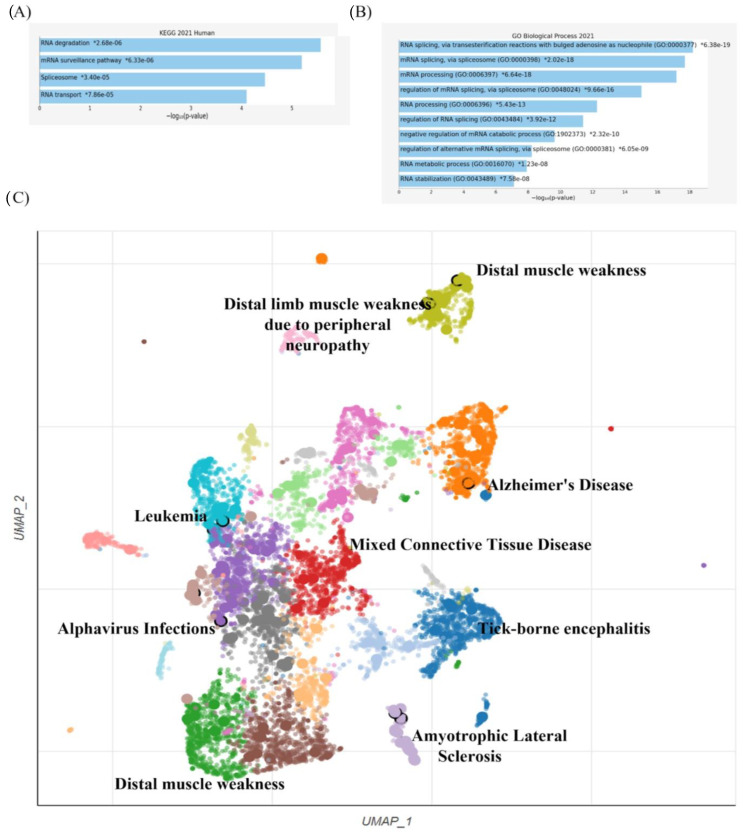
(**A**). The bar chart shows the top 4 enriched terms in KEGG_2021 library, along with their corresponding *p*-values. Colored bars correspond to terms with significant *p*-values (<0.05). An asterisk (*) next to a *p*-value indicates the term also has a significant adjusted *p*-value (<0.05). (**B**) The bar chart of the top 10 enriched terms in GO-biological process library. (**C**) UMAP scatterplot enriched analysis for the DisGeNET database. A library term is represented by each point. The gene set associated with each word had its term frequency-inverse document frequency (TF-IDF) values calculated, and the resulting values were then subjected to UMAP. The first two UMAP dimensions are used to map the terms. Term positions tend to be closer together for gene sets that are more similar. The TF-IDF data are transformed into automatically selected clusters using the Leiden technique. The query is more profoundly enhanced the darker and bigger the point.

## Data Availability

The data that support the findings of this study are available on request from the corresponding author.

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
