# Peer review of "The Continuous Adaptive Challenge Played by Arboviruses: An In Silico Approach to Identify a Possible Interplay between Conserved Viral RNA Sequences and Host RNA Binding Proteins (RBPs)"

_ijms, 2023, doi:10.3390/ijms241311051_

Round 1

Reviewer 1 Report

The information provided is interesting and relevant to the field of the study; however, I would suggest authors to add some information below

- Line4, 6, 145.... no full stop. Check all manuscripts.

-Line 12: delete (;)

-Line 133 Add the names of the viruses used in this study to the Togaviridae, Flaviviridae, and Bunyaviridae families.

-Line 320: Remove the 2.5 Figures section and move the figure to the explain part of each Figure.

-Line 420: In section 4. Materials and Methods, please explain the selection parameters or criteria for virus whole genomes for analysis. 

NA

Author Response

Reviewer #1

The information provided is interesting and relevant to the field of the study: however, I would suggest authors to add some information below

- Line 4, 6, 145.... no full stop. Check all manuscripts.

We checked in the submitted pdf manuscript and found already full stops in line 4, 6 and 145. By the way we checked again the all manuscript to fix any additional errors.

-Line 12: delete (;)

We deleted the punctuation reported.

-Line 133 Add the names of the viruses used in this study to the Togaviridae, Flaviviridae, and Bunyaviridae families.

We added the names of the viruses as suggested

-Line 320: Remove the 2.5 Figures section and move the figure to the explain part of each Figure.

We inserted figures in the appropriate places as suggested by the reviewer.

 -Line 420: In section 4. Materials and Methods, please explain the selection parameters or criteria for virus whole genomes for analysis.

We thank the reviewer for his/her suggestion and explained the used parameters in the manuscript part suggested.

Reviewer 2 Report

Comments on Chetta et al., “The continuous adaptive challenge played by arboviruses: an in silico approach to define relevant molecular interactions with the host”

This is an interesting bioinformatic examination of viral RNA sequences over several families and orders of viruses that infect humans.  The analysis of the sequences reveals some motifs that have the potential to bind to host proteins with known RNA binding preferences, and an effort is made to correlate the potential sequestration or re-localization of host proteins to some pathologies that are associated with infections.  The approach is interesting and potentially powerful, although it still appears quite speculative without experimental confirmation that these interactions actually occur in infected cells.

Title should be more descriptive about what the paper is about, i.e. that the paper proposes that conserved viral RNA elements may sequester host proteins

Abstract is interesting but would benefit from a little more specificity about the nature of the scientific/ bioinformatic approach that is actually used to predict RNA protein interactions.

Line 49-51 and line 109: the term ‘Bunyaviridae’ is no longer used as a family name, rather the Bunyavirales is an Order of viruses.  Likewise, the family names (line 115) have been changed to Phenuivirdae, Hantaviridae, Nairoviridae, etc…

Would be good to state in general terms how the conserved motifs were identified at the beginning of the Results.  Some detail is provided in the Materials and Methods, but it would be useful to know generally what the input dataset was and how it was analyzed, i.e., how did you come up with the Red, Green, Blue consensus sequences.

Likewise, it should be described how the binding/recognition sequences of all of the proposed host proteins were determined and what is the confidence that those proteins actually interact with the different viral RNA sequences during an infection.  The lack of firm experimental verification of the viral RNA/host protein interactions makes the conclusions rather speculative, even if those interactions are consistent with some of the pathological manifestations of viral infections.

The font in Figure 1B is too small to read, and knowing what viruses they represent is important to the interpretation.

Overall, this is quite interesting, although very speculative.  It is ok as a theoretical examination of potential host protein binding to viral RNA sequences, but there is no compelling data that show that these actually happen in cells.  Bolstering the theoretical interactions with empirical observation would dramatically enhance the significance of the manuscript.

Author Response

Reviewer #2

-Title should be more descriptive about what the paper is about, i.e. that the paper proposes that conserved viral RNA elements may sequester host proteins.

We thank the reviewer for his/her suggestion and changed the title in: “The continuous adaptive challenge played by arboviruses: an in silico approach to identify a possible interplay between conserved viral RNA sequences and host RNA binding proteins (RBPs)”.

-Abstract is interesting but would benefit from a little more specificity about the nature of the scientific/ bioinformatic approach that is actually used to predict RNA protein interactions.

According to the suggestion of the reviewer, we improved the abstract. Moreover we elaborated a graphical abstract which we believe could explain more clearly the scientific/ bioinformatic approach proposed.

-Line 49-51 and line 109: the term ‘Bunyaviridae’ is no longer used as a family name, rather the Bunyavirales is an Order of viruses.  Likewise, the family names (line 115) have been changed to Phenuivirdae, Hantaviridae, Nairoviridae, etc…

We changed in the text accordingly.

-Would be good to state in general terms how the conserved motifs were identified at the beginning of the Results.  Some detail is provided in the Materials and Methods, but it would be useful to know generally what the input dataset was and how it was analyzed, i.e., how did you come up with the Red, Green, Blue consensus sequences. Likewise, it should be described how the binding/recognition sequences of all of the proposed host proteins were determined and what is the confidence that those proteins actually interact with the different viral RNA sequences during an infection. 

According to the suggestion of the reviewer, we implemented Materials and methods

-The font in Figure 1B is too small to read, and knowing what viruses they represent is important to the interpretation.

We modified the figure as suggested by the reviewer

-The lack of firm experimental verification of the viral RNA/host protein interactions makes the conclusions rather speculative, even if those interactions are consistent with some of the pathological manifestations of viral infections.

Overall, this is quite interesting, although very speculative.  It is ok as a theoretical examination of potential host protein binding to viral RNA sequences, but there is no compelling data that show that these actually happen in cells.  Bolstering the theoretical interactions with empirical observation would dramatically enhance the significance of the manuscript.

The reviewer 2 properly pointed out in his/her remarks that the proposed investigative pipeline needs experimental validation, however they are beyond the available funds and also beyond the scope of the submitted manuscript. On the other hand, we believe that our findings could  provide a useful tool in further investigations which could focus the attention on conserved sequences that may be used in research and in the development of new diagnostics and therapeutics.